# Fourier Neural Filter as Generic Vision Backbone

## Abstract

Effective information extraction has long been a central challenge in Computer Vision (CV). Transformer- and Mamba-based backbones have significantly advanced this field by providing powerful long-range modeling capability, even though they are initially developed for Natural Language Processing (NLP). Recent progress has highlighted the potential of Fourier Neural Operator (FNO), which, with its favorable quasi-linear complexity and strong global modeling capacity, offers a promising alternative for visual representation learning. However, FNO exhibits a fundamental limitation in capturing local high-frequency patterns due to the over-smoothing effect and bandwidth bottleneck. To address this limitation, we propose Vision Filter (ViF), as a generic backbone for CV, consisting of two complementary components: adaptive modulation for enhancing sensitivity to high-frequency component in the frequency domain, and selective activation for balancing local time-domain and global frequency-domain information flow. Extensive experiments reveal that ViF consistently outperforms prominent variants of Transformer- and Mamba-based backbones across diverse visual tasks, including image classification, object detection, and semantic segmentation. ViF demonstrates lower computational complexity than Transformer-based models and better structural modeling than Mamba-based models, which suffer from spatial disruption due to their directional scanning mechanism. The joint time- and frequency-domain mechanism introduced in ViF may establish a promising paradigm for designing effective visual representation learning, bridging local high-frequency information with global low-frequency information.

## 1 Introduction

Computer Vision (CV) has witnessed remarkable progress in developing architectures capable of extracting meaningful visual information. The evolution from foundational Convolutional Neural Network (CNN) Krizhevsky et al. (2012); Simonyan & Zisserman (2015); He et al. (2016); Liu et al. (2022b) to more complex architectures has been motivated by the essential challenge of balancing computational efficiency with representational capacity Vaswani et al. (2017); Katharopoulos et al. (2020). The introduction of Vision

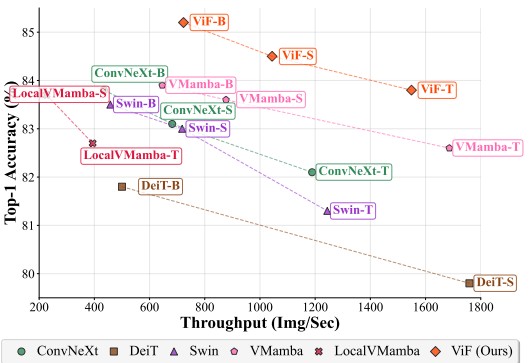

Figure 1: **Model Efficiency Comparison on ImageNet-1k.** For throughput testing, we employ a H100 GPU with a batch size of 128 and an input resolution of 224 × 224.

Transformer (ViT) Dosovitskiy et al. (2020) fundamentally changed visual representation learning by adapting the Transformer backbone from Natural Language Processing (NLP) to CV. By enabling each local patch to dynamically attend to the global context, ViT successfully transcended the inherent local receptive field constraints of traditional convolutional approaches, achieving exceptional model performance across various visual tasks Liu et al. (2021). However, the quadratic computational complexity of Transformer poses significant scalability challenges, particularly when processing high-resolution visual tasks, which has driven researchers to explore alternative

backbones that preserve global modeling capability while achieving superior computational efficiency Touvron et al. (2021); Chu et al. (2021).

The introduction of Mamba has emerged as a compelling solution to address these scalability concerns Gu & Dao (2023); Dao & Gu (2024), sparking considerable research interest and inspiring the development of numerous variants of Vision Mamba (ViM) Zhu et al. (2024); Liu et al. (2024). These approaches have shown promising results across diverse visual tasks, including image restoration Guo et al. (2024) and video understanding Li et al. (2024a). However, these approaches encounter fundamental limitation in preserving the inherent spatial structure of $2D$ visual information, with the principal challenge arising from directional scanning strategy that inevitably lead to spatial disruption Yu & Wang (2024); Han et al. (2024). Recent work Li et al. (2024b) has begun exploring how to construct a more robust scanning mechanism to incorporate spatial-specific inductive biases to improve the representation learning capability of ViM.

Fourier Neural Operator (FNO) Li et al. (2021) offers an alternative paradigm that naturally operates in the $2D$ frequency domain, providing **quasi-linear computational complexity of** $O(N \log N)$ while preserving strong global modeling capacity. Unlike Transformer and Mamba-based models that require converting $2D$ visual representations into $1D$ sequences, FNO directly processes spatial information in its native $2D$ frequency-domain representation, avoiding the associated spatial distortion. However, FNO exhibits fundamental limitations in modeling local high-frequency patterns Liu-Schiaffini et al. (2024) due to the over-smoothing effect and bandwidth bottleneck Rahaman et al. (2019). This inspires the development of more effective Fourier-based backbones.

To address these challenges, we propose **Fourier Neural Filter (FNF)**, a novel nonlinear integral kernel operator that integrates spatial-specific inductive biases directly into the backbone design. Mathematically, FNF extends the standard FNO Li et al. (2021) by introducing an input-dependent kernel function that enables selective activation of local time-domain and global frequency-domain information flow through Hadamard product operations, making it particularly effective for capturing the unique properties of $2D$ visual information. This input-dependent gated global convolution substantially addresses the bandwidth bottleneck by preserving informative mid-/high-frequency components while suppressing redundant ones. On the other hand, to mitigate over-smoothing effect, we incorporate adaptive modulation following complex operation, enabling non-uniform amplification and attenuation of specific frequency bands under stability constraints.

Building upon FNF, we construct **Vision Filter (ViF)** as a generic backbone for CV. Our extensive experiments demonstrate that ViF consistently outperforms prominent variants of both Transformer- and Mamba-based backbones across diverse visual tasks, including image classification on ImageNet-1K Deng et al. (2009), as shown in Fig. 1, object detection on COCO Lin et al. (2014), and semantic segmentation on ADE20K Zhou et al. (2019).

Our contributions are as follows: (1) We propose FNF, the first unified backbone that couples time-domain and frequency-domain analysis, inherently preserving the spatial structure of $2D$ visual representation; (2) We theoretically and empirically demonstrate that our proposed FNF resolves the inherent over-smoothing effect and bandwidth bottleneck of the original FNO; (3) The proposed model ViF achieves state-of-the-art performance on three mainstream visual tasks.

## 2 RELATED WORK

**Vision Transformer**  Building on the success of Vision Transformer (ViT) Dosovitskiy et al. (2020), subsequent developments have focused on making it more efficient and effective through various techniques. These include hierarchical designs like Swin Transformer Liu et al. (2021), PVT Wang et al. (2021), and NAT Hassani et al. (2023), hybrid approaches combining CNN with Transformers like CMT Guo et al. (2022), CrossViT Chen et al. (2021), MaxVit Tu et al. (2022), and FasterViT Hatamizadeh et al. (2023), and large-scale self-supervised pre-training models like MAE He et al. (2022) and BEIT Bao et al. (2022). These innovations have collectively established ViT as a fundamental architecture for diverse visual tasks.

**Vision Mamba**  Recent work on Vision Mamba (ViM) is aiming to overcome the fundamental limitations of its directional scanning strategy for $2D$ visual information processing, including bi-directional Zhu et al. (2024) and quad-directional Liu et al. (2024) scanning, and other approaches capable of balancing both local and global information extraction Pei et al. (2024); Huang et al.

(2024); Xiao et al. (2024). These advances collectively improve representation learning and spatial understanding of ViM by addressing the inherent challenges of applying autoregressive models to $2D$ visual information while maintaining computational efficiency.

**Fourier Transform for Vision**   Previous work has successfully integrated Fourier transform into deep learning system Lee-Thorp et al. (2021). GFNet Rao et al. (2021) achieves competitive performance with logarithmic linear complexity by replacing the self-attention mechanism in the ViT backbone with $2D$ discrete Fourier transform and learnable global filter. FourCastNet Pathak et al. (2022); Kurth et al. (2023), developed based on AFNO Guibas et al. (2022), generates one-week global weather forecasting within 2 seconds—several orders of magnitude faster than traditional numerical weather forecasting models Pathak et al. (2022). Recent extensions include SFNO Bonev et al. (2023), which incorporates spherical harmonic transforms into atmospheric modeling to enable stable year-round weather forecasting on spherical geometry Bonev et al. (2025).

## 3 METHODOLOGY

In this section, we theoretically analyze the limitations of Fourier Neural Operator (FNO) and introduce the fundamentals of our proposed Fourier Neural Filter (FNF).

### 3.1 LIMITATIONS OF FOURIER NEURAL OPERATOR (FNO)

**Proposition 1 (Bandwidth Bottleneck.)**   Consider a periodic functions $v$ expanded in a Fourier series. Let $P_K$ denote the projection onto Fourier modes $\{|k| \leqslant K\}$. Any FNO layer $F_K(v)$ with fixed bandwidth $K$ depends only on $P_K$. If $v$ is non-bandlimited, and the operator $\mathcal{T}$ is not strictly low-pass, leading to an irreducible *truncation error* in the frequency domain:

$$\inf_{F_K} \left\| F_K(v) - \mathcal{T}(v) \right\| \geqslant \left\| P_K^\perp \mathcal{T}(v) \right\|. \tag{1}$$

*Proof sketch.* FNO applies a fixed spectral map on $\{|k| \leqslant K\}$ and discards $\{|k| > K\}$. Therefore, two inputs with identical $P_K$ cannot be distinguished. The error lower bound follows from orthogonal decomposition into $P_K$ and $P_K^\perp$ components.

**Proposition 2 (Over-smoothing Effect.)**   Let $M_\ell(k)$ be the per-layer spectral multipliers on $\{|k| \leqslant K\}$. If there exists $\rho \in (0, 1)$ and $k_0 \leqslant K$ such that $|M_\ell(k)| \leqslant \rho$ for all $|k| \geqslant k_0$ and all layers $\ell$, then the overall frequency response $H_L(k) = \prod_{\ell=1}^{L} M_\ell(k)$ can satisfy $|H_L(k)| \leqslant \rho^L \to 0$ on $\{|k| \geqslant k_0\}$ as $L \to \infty$, leading to an *over-smoothing* spatial representation.

*Proof sketch.* Multiplicative contraction on the mid-/high-frequency modes accumulates exponentially with depth; coupled with the hard truncation outside $\{|k| \leqslant K\}$, the output energy is concentrated in the low-frequency modes, while the high-frequency modes are progressively suppressed.

### 3.2 FUNDAMENTALS OF FOURIER NEURAL FILTER (FNF)

While FNO Li et al. (2021) has demonstrated remarkable effectiveness in modeling complex dynamic systems and solving partial differential equations through fixed integral kernel, our proposed FNF (Fig. 2) makes a critical leap forward: introducing an input-dependent integral kernel that can allow for adaptive and dynamic information flow between the time and frequency domains, thereby constructing a unified time-frequency representation space. Intuitively, if FNO applies a fixed lens to process all input signals, then FNF continuously adjusts the lens based on the preceding scene, achieving more detailed information extraction and more robust pattern recognition. We analyze the theoretical underpinnings of FNF by examining

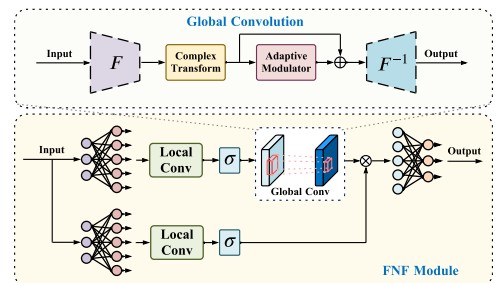

Figure 2: **Schematic diagram of our proposed Fourier Neural Filter (FNF) backbone.**

integral kernel, global convolution, selective activation, complex transform, and adaptive modulation.

### 3.2.1 INTEGRAL KERNEL

**Definition 1**  FNO is defined via a fixed integral kernel operator:

$$(Kv)(x) = \int_D \kappa(x, y)v(y)\, dy, \tag{2}$$

where $\kappa : D \times D \to \mathbb{R}$ is the kernel function and $v : D \to \mathbb{R}$ is the input function. Through the Fourier transform, FNO can be formulated in the frequency domain as:

$$(Kv)(x) = \mathcal{F}^{-1}(R_\phi \cdot \mathcal{F}(v))(x), \tag{3}$$

where $R_\phi = \mathcal{F}(\kappa)$ denotes the parameterized frequency-domain kernel.

**Definition 2**  FNF can be defined through an adaptive integral kernel operator:

$$(Kv)(x) = \int_D \kappa(x, y; v)v(y)\, dy, \tag{4}$$

where $\kappa(x, y; v)$ is the input-dependent kernel function. In the implementation, FNF can also be formulated as:

$$(Kv)(x) = T(G(v) \odot P(v))(x), \tag{5}$$

$$P(v)(x) = \mathcal{F}^{-1}(R_\phi \cdot \mathcal{F}(H(v)))(x), \tag{6}$$

where $G(v)$, $H(v)$, and $T(v)$ denote the linear transform used for expansion or compression, and $\odot$ is the Hadamard product operation.

**Remark 1**  The fundamental distinction between FNO and FNF lies in their kernel functions: FNO employs a fixed kernel $\kappa(x, y)$, whereas FNF applies an input-dependent kernel $\kappa(x, y; v)$, enabling adaptive information flow modulation between time-domain and frequency-domain, constructing a unified time-frequency representation space.

### 3.2.2 GLOBAL CONVOLUTION

**Definition 3**  When the kernel function $\kappa(x, y) = \kappa(x - y)$ exhibits translation invariance, the fixed integral kernel operator in FNO reduces to a global convolution Li et al. (2021):

$$(Kv)(x) = \int_D \kappa(x - y)v(y)\, dy = (\kappa * v)(x). \tag{7}$$

**Definition 4**  Similarly, when the kernel function $\kappa(x, y; v) = \kappa(x - y; v)$ maintains translation invariance, the adaptive integral kernel operator in FNF becomes a gated global convolution:

$$(Kv)(x) = \int_D \tilde{\kappa}(x - y; v)v(y)\, dy = (\tilde{\kappa}(\cdot; v) * v)(x). \tag{8}$$

**Remark 2**  Translation invariance enables efficient computation of integral operator through Fourier transform in both FNO and FNF. Beyond this shared efficiency, the gated global convolution in FNF significantly enhances representation capacity by employing an input-dependent kernel $\tilde{\kappa}(\cdot; v)$, which adaptively modulates filtering behavior while preserving computational efficiency.

### 3.2.3 SELECTIVE ACTIVATION

**Definition 5**  The selective activation operates an element-wise multiplication in the time domain; in the frequency domain, this operation is mathematically equivalent to the convolution operation between $G(v)(x)$ and $P(v)(x)$:

$$\mathcal{F}(G(v) \odot P(v))(\omega) = (\hat{G}(v) * \hat{P}(v))(\omega). \tag{9}$$

This formula can be viewed as approximate magnitude modulation and phase addition when the signal $G(v)$ is relatively smooth or narrow:

$$(G(v) \odot P(v))_i \approx |G(v)_i| \cdot |P(v)_i| \cdot e^{i(\theta_{G(v)_i} + \theta_{P(v)_i})}, \tag{10}$$

where $|G(v)_i|$ and $|P(v)i|$ represent magnitudes, and $\theta G(v)i$ and $\theta P(v)_i$ represent phases.

**Remark 3** This formulation reveals how selective activation effectively achieves joint time–frequency modulation: it enhances informative mid-/high-frequency components while suppressing redundant low-frequency ones on the magnitude side, and simultaneously provides flexible alignment on the phase side. This design alleviates the well-known over-smoothing effect and bandwidth bottleneck Rahaman et al. (2019) of FNO and improves the representation learning capability.

### 3.2.4 COMPLEX TRANSFORM

**Definition 6** The complex transform operates on the complex-valued input $z = z_r + iz_i$ with complex weights $W = W_r + iW_i$ and biases $b = b_r + ib_i$:

$$L(z) = (W_r z_r - W_i z_i + b_r) + i(W_r z_i + W_i z_r + b_i). \tag{11}$$

**Remark 4** To reduce the parameter count, we adopt the block-diagonal structure for the weights Guibas et al. (2022) and implement two complex transform layers equipped with the GELU activation function Hendrycks & Gimpel (2016).

### 3.2.5 ADAPTIVE MODULATION

**Definition 7** The adaptive modulation operates through an amplitude-sensitive weighting function to achieve frequency balancing Liu & Tang (2025):

$$\mathcal{M}(z) = z \odot [\beta \cdot \|z\|^\alpha], \tag{12}$$

where $\|z\|$ represent the magnitude of complex-valued input $z$, and $\alpha, \beta$ are learnable parameters, $\odot$ is the Hadamard product operation.

**Remark 5** When $\alpha < 1$, the power-law weighting compresses the dynamic range between frequency components, effectively attenuating dominant low-frequency components while relatively enhancing weak high-frequency components. On the other hand, the adaptive parameter $\beta$ provides global scaling control to achieve optimal frequency balance.

## 4 MODEL

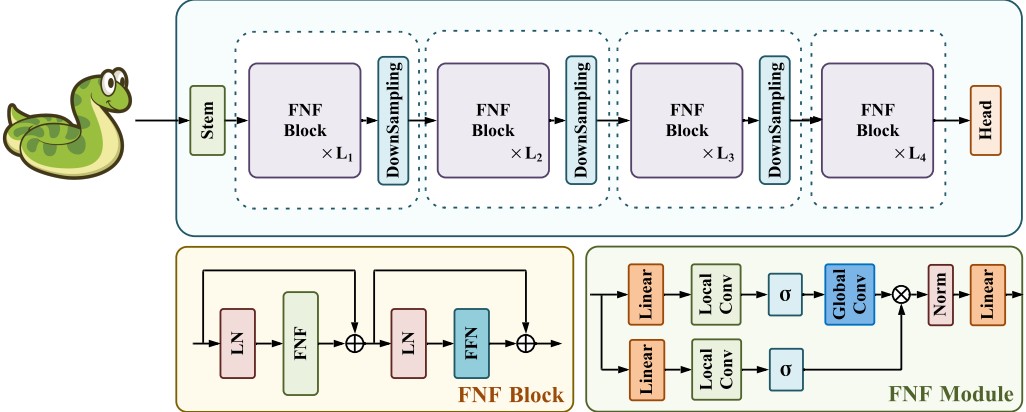

Figure 3: **Schematic diagram of our proposed Vision Filter (ViF) architecture.** Architecture details can be found in the Appendix.

**Overall Architecture** Our ViF model is structured into four hierarchical stages, as shown in Fig. 3, mirroring the design principles of established vision backbones in previous works Liu et al. (2021; 2022b; 2024). Specifically, an input image $\mathbf{I} \in \mathbb{R}^{H \times W \times 3}$ is initially processed through an overlapped stem layer to obtain a $2D$ feature map with dimension of $\frac{H}{4} \times \frac{W}{4} \times C$. This feature map is subsequently fed into four successive stages, where each stage comprises multiple ViF blocks followed by down-sampling layer with reduction factor of 2 (excluding the final stage). The head layer processes the feature map to obtain the spatial representation tailored for specific downstream tasks. More details can be found in the Appendix.

**Block Design**   The ViF block serves as the fundamental construction unit of our architecture, including the FNF and Feed-Forward Network (FFN) modules with residual skip connection He et al. (2016), as shown in the lower-left corner of Fig. 3. Our FNF module, illustrated in the bottom lower-right corner of Fig. 3, has two branches: one branch contains a local convolution and a global convolution enabling to capture effective spatial information through progressive learning from local to global representation, the other branch contains a local convolution enabling to achieve effective fusion of global frequency-domain information and local time-domain information through Hadamard product operation. Additionally, FFN module is added subsequent to the FNF module to promote information flow interaction across channels and to maintain alignment with the settings of classical ViTs. Furthermore, Local Perception Unit (LPU) Guo et al. (2022) is employed before both the FNF and FFN module to incorporate local inductive biases.

## 5 EXPERIMENT

In this section, to validate the effectiveness of our proposed ViF, we conduct extensive experiments on a variety of visual tasks, including image classification, object detection, and semantic segmentation. Following the previous works Liu et al. (2021; 2024), we train three variants of ViF, called ViF-T, ViF-S and ViF-B, as shown in Tab. 1.

Table 1: **Model Description of ViF variants.**

| Models | Blocks | Channels | Heads |
|---|---|---|---|
| ViF-Tiny | [2, 4, 8, 4] | [64, 128, 256, 512] | [2, 4, 8, 16] |
| ViF-Small | [2, 5, 19, 5] | [64, 128, 256, 512] | [2, 4, 8, 16] |
| ViF-Base | [2, 5, 19, 5] | [96, 192, 384, 768] | [3, 6, 12, 24] |

### 5.1 IMAGE CLASSIFICATION ON IMAGENET-1K

**Settings**   We conduct a comprehensive evaluation of ViF on image classification using ImageNet-1K dataset Deng et al. (2009). Our experimental setup follows the configurations established in the previous works Liu et al. (2021; 2024), with complete implementation details provided in the Appendix. We compare our model with other state-of-the-art models, including CNN-based models (RegNetY Radosavovic et al. (2020), ConvNeXt Liu et al. (2022b), and MambaOut Yu & Wang (2024)), Transformer-based models (ViT Dosovitskiy et al. (2020), DeiT Touvron et al. (2021), Swin Liu et al. (2021), SwinV2 Liu et al. (2022a), Twins Chu et al. (2021), and NAT Hassani et al. (2023)), Mamba-based models (ViM Zhu et al. (2024), VMamba Liu et al. (2024), LocalVMamba Huang et al. (2024), EfficientVMamba Pei et al. (2024), and MambaVision Hatamizadeh & Kautz (2025)), and Fourier-based models (GFNet and GFNetV2 Rao et al. (2021)).

**Results**   The experimental results on ImageNet-1K image classification are reported in Tab. 2. Compared to Transformer-based models, ViF-T exceeds Swin-T by 2.3% and NAT-T by 0.6%. In comparison with Mamba-based models, ViF-T outperforms VMamba-T by 1.3% and LocalVMamba-T by 1.1%. Among Fourier-based models, ViF demonstrates substantial improvements over existing approaches: ViF-T surpasses GFNet-S by 3.8% and GFNetV2-B by 1.7%, showcasing the superiority of our proposed architecture design. For larger variants, ViF-S and ViF-B achieve the accuracy of 84.5% and 85.2%, respectively, significantly outperforming GFNetV2-S by 2.8% and GFNetV2-B by 3.1%, while surpassing NAT-S and NAT-B by 1.5% and 0.9%, and VMamba-S and VMamba-B by 0.9% and 1.3%. These comprehensive results demonstrate that ViF achieves outstanding model performance across different model sizes while maintaining competitive computational efficiency.

### 5.2 OBJECT DETECTION ON COCO

**Settings**   We conduct a comprehensive evaluation of ViF on object detection using COCO 2017 dataset Deng et al. (2009) and MMDetection library. We adopt Mask R-CNN He et al. (2017) as detector, and apply the pre-trained ViF-T/S/B as backbone. Following the previous work Liu et al. (2021; 2024), we fine-tune the pre-trained models on the COCO dataset for single-scale training ($1\times$ schedule) and multi-scale training ($3\times$ schedule).

Table 2: **Comparison of image classification performance on ImageNet-1K.**

| Architecture | Method | Image Size | Params (M) | FLOPs (G) | Top-1 (%) |
|---|---|---|---|---|---|
| CNN | RegNetY-4G | $224^2$ | 21 | 4.0 | 80.0 |
| | RegNetY-8G | $224^2$ | 39 | 8.0 | 81.7 |
| | RegNetY-16G | $224^2$ | 84 | 16.0 | 82.9 |
| | ConvNeXt-T | $224^2$ | 29 | 4.5 | 82.1 |
| | ConvNeXt-S | $224^2$ | 50 | 8.7 | 83.1 |
| | ConvNeXt-B | $224^2$ | 89 | 15.4 | 83.8 |
| | MambaOut-T | $224^2$ | 27 | 4.5 | 82.7 |
| | MambaOut-S | $224^2$ | 48 | 9.0 | 84.1 |
| | MambaOut-B | $224^2$ | 85 | 15.8 | 84.2 |
| Transformer | ViT-B/16 | $384^2$ | 86 | 55.4 | 77.9 |
| | DeiT-S | $224^2$ | 22 | 4.6 | 79.8 |
| | DeiT-B | $224^2$ | 87 | 16.9 | 81.8 |
| | Swin-T | $224^2$ | 28 | 4.5 | 81.3 |
| | Swin-S | $224^2$ | 50 | 8.7 | 83.0 |
| | Swin-B | $224^2$ | 88 | 15.4 | 83.5 |
| | SwinV2-T | $256^2$ | 28 | 4.8 | 82.7 |
| | SwinV2-S | $256^2$ | 50 | 8.5 | 83.5 |
| | SwinV2-B | $256^2$ | 88 | 15.1 | 84.6 |
| | Twins-S | $224^2$ | 24 | 2.8 | 81.7 |
| | Twins-B | $224^2$ | 56 | 8.3 | 83.1 |
| | NAT-T | $224^2$ | 28 | 4.3 | 83.2 |
| | NAT-S | $224^2$ | 51 | 7.8 | 83.0 |
| | NAT-B | $224^2$ | 90 | 13.7 | 84.3 |
| Mamba | ViM-S/16 | $224^2$ | 26 | 5.1 | 80.3 |
| | VMamba-T | $224^2$ | 30 | 4.9 | 82.6 |
| | VMamba-S | $224^2$ | 50 | 8.7 | 83.6 |
| | VMamba-B | $224^2$ | 89 | 15.4 | 83.9 |
| | LocalVMamba-T | $224^2$ | 26 | 5.7 | 82.7 |
| | LocalVMamba-S | $224^2$ | 50 | 11.4 | 83.7 |
| | EfficientVMamba-S | $224^2$ | 11 | 1.3 | 78.7 |
| | EfficientVMamba-B | $224^2$ | 33 | 4.0 | 81.8 |
| | MambaVision-T | $224^2$ | 32 | 4.4 | 82.3 |
| | MambaVision-S | $224^2$ | 50 | 7.5 | 83.3 |
| | MambaVision-B | $224^2$ | 98 | 15.0 | 84.2 |
| Fourier | GFNet-S | $224^2$ | 25 | 4.5 | 80.0 |
| | GFNet-B | $224^2$ | 43 | 7.9 | 80.7 |
| | GFNetV2-S | $384^2$ | 28 | 13.2 | 81.7 |
| | GFNetV2-B | $384^2$ | 47 | 23.3 | 82.1 |
| | ViF-T | $224^2$ | 29 | 5.1 | **83.8** |
| | ViF-S | $224^2$ | 45 | 7.8 | **84.5** |
| | ViF-B | $224^2$ | 96 | 16.7 | **85.2** |

**Results**   The experimental results on COCO object detection are reported in Tab. 3. Under the single-scale training, ViF-T achieves a box mAP of 47.7 and a mask mAP of 43.0, surpassing Swin-T by 5.0 and 3.7, respectively, while using comparable computational costs (48M parameters and 272G FLOPs vs. 48M parameters and 267G FLOPs). Compared to VMamba-T, ViF-T shows competitive performance with improvement of 0.4 in box mAP and 0.3 in mask mAP, while maintaining similar computational costs. For larger variants, ViF-S achieves 49.1 box mAP and 44.0 mask mAP, outperforming VMamba-S by 0.4 and 0.3, respectively, with reduced computational costs (64M parameters and 328G FLOPs vs. 70M parameters and 349G FLOPs). Under the multi-scale training schedule, these performance advantages are maintained and even enhanced. ViF-T achieves 48.9 box mAP and 43.4 mask mAP, while ViF-S reaches the highest performance with 50.1 box mAP and 44.4 mask mAP, outperforming VMamba-S by 0.2 and 0.2, respectively. These comprehensive results demonstrate the effectiveness and robustness of ViF architectures for dense prediction tasks.

Table 3: **Comparison of object detection performance on COCO with Mask R-CNN He et al. (2017) detector.** FLOPs are calculated with input resolution of $1280 \times 800$.

| Backbone | $AP^b\uparrow$ | $AP^b_{50}\uparrow$ | $AP^b_{75}\uparrow$ | $AP^m\uparrow$ | $AP^m_{50}\uparrow$ | $AP^m_{75}\uparrow$ | Params | FLOPs |
|---|---|---|---|---|---|---|---|---|
| **Mask R-CNN $1\times$ schedule** | | | | | | | | |
| ResNet-50 | 38.2 | 58.8 | 41.4 | 34.7 | 55.7 | 37.2 | 44M | 260G |
| Swin-T | 42.7 | 65.2 | 46.8 | 39.3 | 62.2 | 42.2 | 48M | 267G |
| ConvNeXt-T | 44.2 | 66.6 | 48.3 | 40.1 | 63.3 | 42.8 | 48M | 262G |
| PVTv2-B2 | 45.3 | 66.1 | 49.6 | 41.2 | 64.2 | 44.4 | 45M | 309G |
| ViT-Adapter-S | 44.7 | 65.8 | 48.3 | 39.9 | 62.5 | 42.8 | 48M | 403G |
| MambaOut-T | 45.1 | 67.3 | 49.6 | 41.0 | 64.1 | 44.1 | 43M | 262G |
| VMamba-T | 47.3 | 69.3 | 52.0 | 42.7 | 66.4 | 45.9 | 50M | 271G |
| LocalVMamba-T | 46.7 | 68.7 | 50.8 | 42.2 | 65.7 | 45.5 | 45M | 291G |
| ViF-T | **47.7** | **70.0** | **52.1** | **43.0** | **66.7** | **46.5** | 48M | 272G |
| ResNet-101 | 38.2 | 58.8 | 41.4 | 34.7 | 55.7 | 37.2 | 63M | 336G |
| Swin-S | 44.8 | 68.6 | 49.4 | 40.9 | 65.3 | 44.2 | 69M | 354G |
| ConvNeXt-S | 45.4 | 67.9 | 50.0 | 41.8 | 65.2 | 45.1 | 70M | 348G |
| PVTv2-B3 | 47.0 | 68.1 | 51.7 | 42.5 | 65.2 | 45.7 | 63M | 397G |
| MambaOut-S | 47.4 | 69.1 | 52.4 | 42.7 | 66.1 | 46.2 | 65M | 354G |
| VMamba-S | 48.7 | 70.0 | 53.4 | 43.7 | 67.3 | 47.0 | 70M | 349G |
| LocalVMamba-S | 48.4 | 69.9 | 52.7 | 43.2 | 66.7 | 46.5 | 69M | 414G |
| ViF-S | **49.1** | **70.4** | **53.5** | **44.0** | **67.6** | **47.5** | 64M | 328G |
| Swin-B | 46.9 | - | - | 42.3 | 66.3 | 46.0 | 88M | 496G |
| ConvNeXt-B | 47.0 | 69.4 | 51.7 | 42.7 | 66.3 | 46.0 | 107M | 486G |
| PVTv2-B5 | 47.4 | 68.6 | 51.9 | 42.5 | 65.7 | 46.0 | 102M | 557G |
| ViT-Adapter-B | 47.0 | 68.2 | 51.4 | 41.8 | 65.1 | 44.9 | 102M | 557G |
| MambaOut-B | 47.4 | 69.3 | 52.2 | 43.0 | 66.4 | 46.3 | 100M | 495G |
| VMamba-B | 49.2 | 71.4 | 54.0 | 44.1 | 68.3 | 47.7 | 108M | 485G |
| ViF-B | **50.1** | **71.3** | **54.8** | **44.6** | **68.5** | **48.1** | 120M | 517G |

| Backbone | $AP^b\uparrow$ | $AP^b_{50}\uparrow$ | $AP^b_{75}\uparrow$ | $AP^m\uparrow$ | $AP^m_{50}\uparrow$ | $AP^m_{75}\uparrow$ | #Param. | FLOPs |
|---|---|---|---|---|---|---|---|---|
| **Mask R-CNN $3\times$ MS schedule** | | | | | | | | |
| Swin-T | 46.0 | 68.1 | 50.3 | 41.6 | 65.1 | 44.9 | 48M | 267G |
| ConvNeXt-T | 46.2 | 67.9 | 50.8 | 41.7 | 65.0 | 44.9 | 48M | 262G |
| NAT-T | 47.7 | 69.0 | 52.6 | 42.6 | 66.1 | 45.9 | 48M | 258G |
| VMamba-T | 48.8 | 70.4 | 53.5 | 43.7 | 67.4 | 47.0 | 50M | 271G |
| LocalVMamba-T | 48.7 | 70.1 | 53.0 | 43.4 | 67.0 | 46.4 | 45M | 291G |
| ViF-T | **48.9** | **70.3** | **53.6** | **43.4** | **67.5** | **46.5** | 48M | 272G |
| Swin-S | 48.2 | 69.8 | 52.8 | 43.2 | 67.0 | 46.1 | 69M | 354G |
| ConvNeXt-S | 47.9 | 70.0 | 52.7 | 42.9 | 66.9 | 46.2 | 70M | 348G |
| NAT-S | 48.4 | 69.8 | 53.2 | 43.2 | 66.9 | 46.5 | 70M | 330G |
| VMamba-S | 49.9 | 70.9 | 54.7 | 44.2 | 68.2 | 47.7 | 70M | 349G |
| LocalVMamba-S | 49.9 | 70.5 | 54.4 | 44.1 | 67.8 | 47.4 | 69M | 414G |
| ViF-S | **50.1** | **71.4** | **54.9** | **44.4** | **68.3** | **47.9** | 64M | 328G |

## 5.3 SEMANTIC SEGMENTATION ON ADE20K

**Settings**   We conduct a comprehensive evaluation of ViF on semantic segmentation using ADE20K dataset Zhou et al. (2019) and MMSegmenation toolkit. We adopt UPerNet Xiao et al. (2018) as segmentor, and apply pre-trained ViF-T/S/B as backbone. Consistent with the previous work Liu et al. (2021; 2024), we fine-tune the pre-trained models on the ADE20K dataset for both both single-scale and multi-scale testing.

**Results**   The experimental results on ADE20K semantic segmentation are reported in Tab. 4. Under the single-scale testing, ViF-T achieves a single-scale mIoU of 48.7 and a multi-scale mIoU of 49.6, representing significant improvements of 1.6 mIoU over NAT-T and 0.7 mIoU over VMamba-T, respectively. Under multi-scale testing, ViF-T maintains its competitive advantage with im-

Table 4: **Comparison of semantic segmentation on ADE20K with UPerNet Xiao et al. (2018) segmentor.** FLOPs are calculated with input resolution of $512 \times 2048$.

| Method | Crop size | mIoU (SS) ↑ | mIoU (MS) ↑ | Params. | FLOPs |
|---|---|---|---|---|---|
| DeiT-S + MLN | $512^2$ | 43.1 | 43.8 | 58M | 1217G |
| Swin-T | $512^2$ | 44.4 | 45.8 | 60M | 945G |
| ConvNeXt-T | $512^2$ | 46.0 | 46.7 | 60M | 939G |
| NAT-T | $512^2$ | 47.1 | 48.4 | 58M | 934G |
| MambaOut-T | $512^2$ | 47.4 | 48.6 | 54M | 938G |
| VMamba-T | $512^2$ | 48.0 | 48.8 | 62M | 949G |
| LocalVMamba-T | $512^2$ | 47.9 | 49.1 | 57M | 970G |
| ViF-T | $512^2$ | **48.7** | **49.6** | 58M | 948G |
| DeiT-B + MLN | $512^2$ | 45.5 | 47.2 | 144M | 2007G |
| Swin-S | $512^2$ | 47.6 | 49.5 | 81M | 1039G |
| ConvNeXt-S | $512^2$ | 48.7 | 49.6 | 82M | 1027G |
| NAT-S | $512^2$ | 48.0 | 49.5 | 82M | 1010G |
| MambaOut-S | $512^2$ | 49.5 | 50.6 | 76M | 1032G |
| VMamba-S | $512^2$ | 50.6 | 51.2 | 82M | 1028G |
| LocalVMamba-S | $512^2$ | 50.0 | 51.0 | 81M | 1095G |
| ViF-S | $512^2$ | **50.5** | **51.3** | 76M | 1009G |
| Swin-B | $512^2$ | 48.1 | 49.7 | 121M | 1188G |
| ConvNeXt-B | $512^2$ | 49.1 | 49.9 | 122M | 1170G |
| NAT-B | $512^2$ | 48.5 | 49.7 | 123M | 1137G |
| MambaOut-B | $512^2$ | 49.6 | 51.0 | 112M | 1178G |
| VMamba-B | $512^2$ | 51.0 | 51.6 | 122M | 1170G |
| ViF-B | $512^2$ | **51.3** | **52.3** | 131M | 1200G |

provements of 1.2 mIoU over NAT-T and 0.8 mIoU over VMamba-T. For larger variants, ViF-S shows superior performance with 50.5 single-scale mIoU and 51.3 multi-scale mIoU, outperforming VMamba-S while using fewer computational costs (76M parameters and 1009G FLOPs vs. 82M parameters and 1028G FLOPs). Notably, ViF-B achieve a single-scale mIoU of 51.3 and multi-scale mIoU of 52.3, surpassing VMamba-B by 0.3 and 0.7, respectively.

**Ablation Study**  To validate the effectiveness of each component in our model, we conduct a comprehensive ablation study, as shown in Tab. 5. Removing LC-1 drops accuracy to 83.6% and removing LC-2 further decreases accuracy to 83.4%, both showing their importance. Eliminating adaptive modulation (AM) leads

Table 5: **Ablation study.** Our ViF-T model is highlighted.

| Model | Top-1 | Params(M) | FLOPs(G) | Throughput |
|---|---|---|---|---|
| w/o LC-1 | 83.6 | 28 | 5.0 | 1585 |
| w/o LC-2 | 83.4 | 28 | 5.0 | 1589 |
| w/o AM | 83.5 | 29 | 5.1 | 1667 |
| w/o SA | 83.1 | 25 | 4.6 | 1689 |
| ViF-T | 83.8 | 29 | 5.1 | 1549 |

to 83.5% accuracy, while removing selective activation (SA) has the largest impact, with accuracy dropping to 83.3%. These results demonstrate the significant impact of each component on model performance, with SA proving most critical for maintaining accuracy.

# 6 CONCLUSION

**Limitations**  While our ViF model outperforms baselines on ImageNet-1K, three key limitations exist: (1) marginal performance gains compared to other ViM models on downstream tasks, (2) significant performance gap against ViT variants on downstream tasks Fan et al. (2024); Shi (2024), and (3) lack of scalability evaluation on larger models and datasets (e.g., ImageNet-22K).

**Broader Impact**  Our ViF model offers significant potential benefits for efficient visual representation learning. However, potential risks include accessibility barriers due to frequency-domain operations and possible perpetuation of biases present in training data. We encourage responsible deployment and ongoing research to address these considerations.

## 7 ETHICS STATEMENT

This work does not involve human subjects, does not raise concerns regarding data privacy, bias, fairness, or potential harmful applications, and does not present conflicts of interest or legal compliance issues. The research methodology and findings do not pose ethical concerns that require additional consideration beyond standard academic practices.

## 8 REPRODUCIBILITY STATEMENT

To ensure reproducibility of our results, we provide the following resources: (1) complete implementation details and hyperparameters are described in Section 5 and Appendix C; (2) all datasets used in our experiments are publicly available and properly cited with access information provided in Section 5; (3) theoretical proofs and derivations are included in Section 3; and (5) source code will be made available upon publication to facilitate replication of our experimental results.

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
