## A LLM USAGE STATEMENT

Large language models were used solely as general-purpose writing assistance tools to aid in polishing the manuscript text and improving clarity of expression. LLMs did not contribute to research ideation, methodology development, experimental design, data analysis, or the generation of scientific insights presented in this work.

## B VISUALIZATIONS

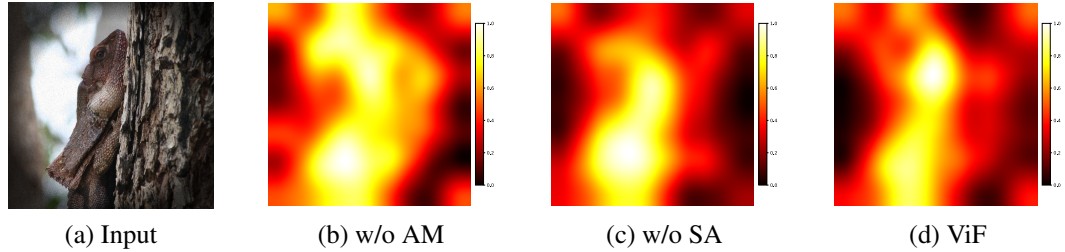

(a) Input          (b) w/o AM          (c) w/o SA          (d) ViF

Figure A1: **Visualization of ablation study for adaptive modulation (AM) and selective activation (SA).**

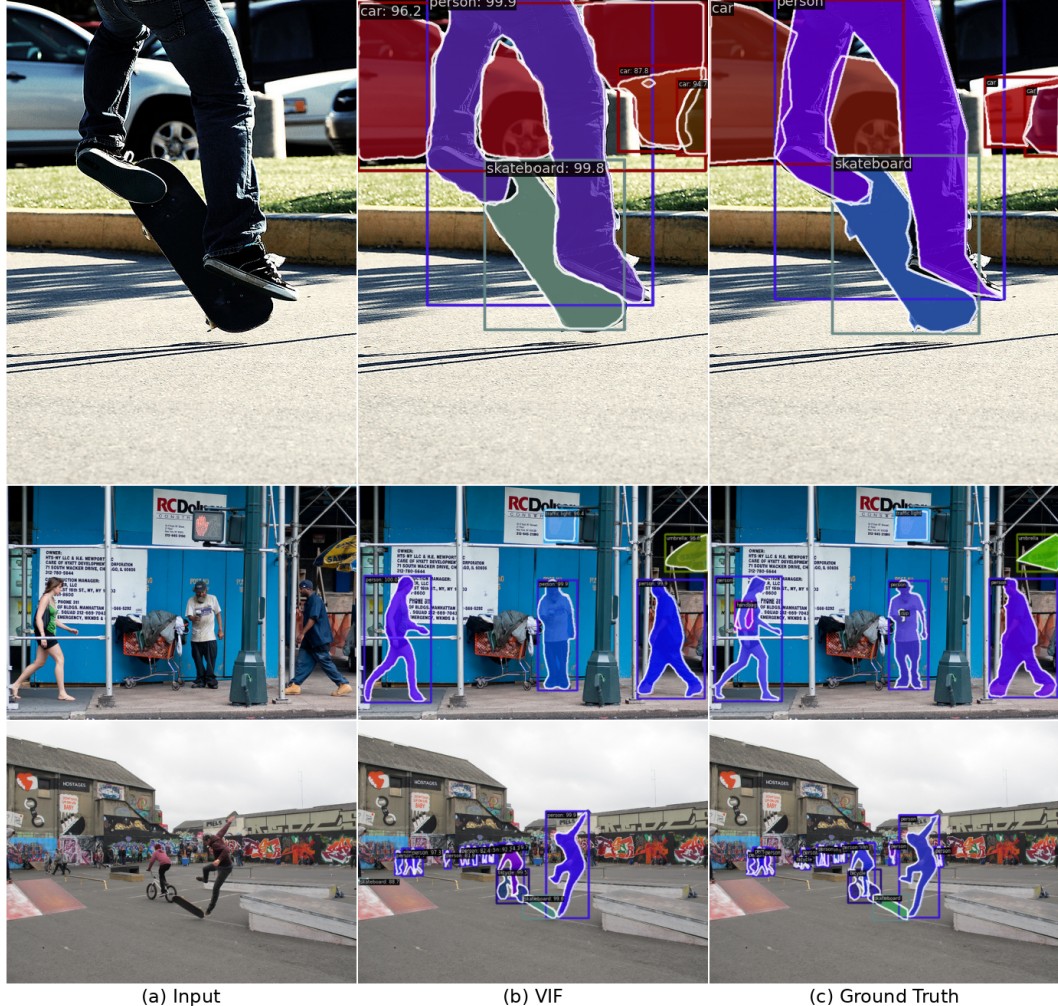

(a) Input          (b) VIF          (c) Ground Truth

Figure A2: **Visualization of object detection.**

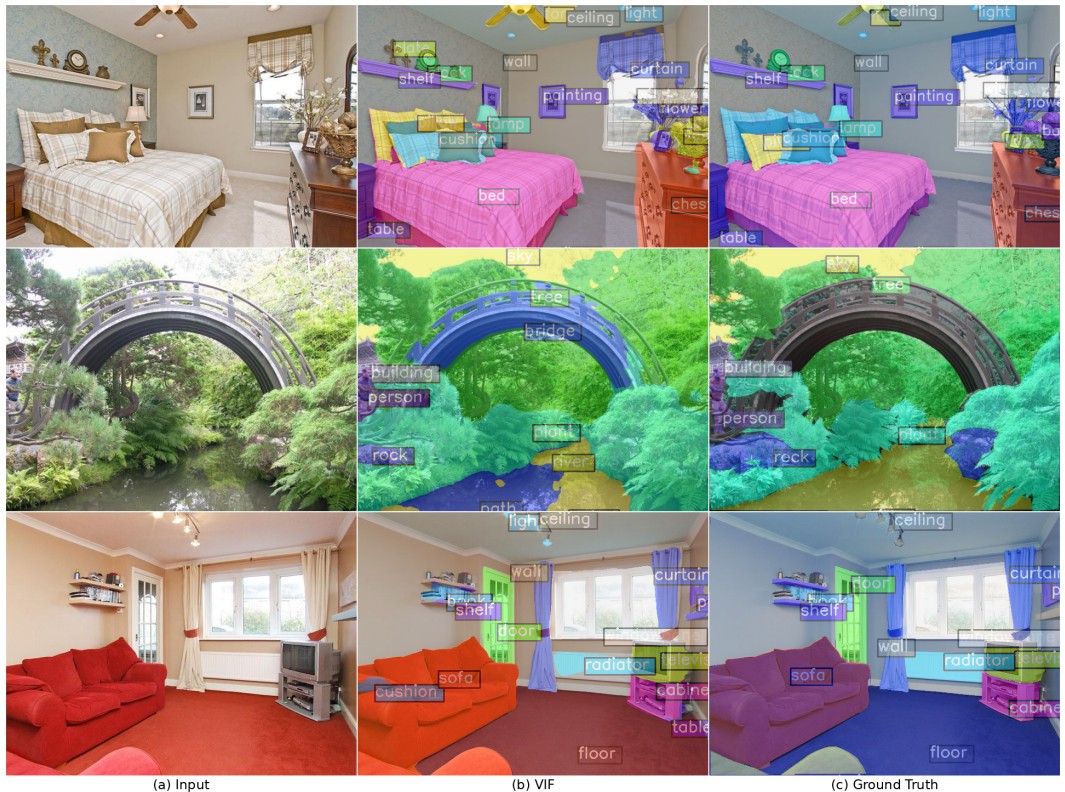

Figure A3: **Visualization of semantic segmentation.**

# C  DATASET DETAILS

**ImageNet-1K for image classification.** The ImageNet-1K dataset (Deng et al., 2009) comprises 1280k images for training and 50k images for validation, with 1,000 categories. For a fair comparison, we train our models under the same settings as Swin Transformer (Liu et al., 2021). Specifically, we employ AdamW optimizer to train our models from scratch for 300 epochs. The initial learning rate is set to $4 \times 10^{-3}$. We apply a cosine learning rate decay schedule with a linear warm-up of 20 epochs and a weight decay of 0.05. The batch size is set to 2048 for ViF-T and ViF-S, and 1024 for ViF-B. The drop path rate is set to 0.2 for ViF-T, 0.3 for ViF-S and 0.5 for ViF-B. MESA (Du et al., 2022) is used to prevent overfitting. Data augmentation techniques including RandAugment (Cubuk et al., 2020), Mixup (Zhang et al., 2018), CutMix (Yun et al., 2019), and random erasing (Zhong et al., 2020) are employed during training.

**COCO for object detection.** The COCO dataset (Lin et al., 2014) comprises 118K images for training and 5K images for validation, with 80 object categories. We follow the standard $1\times$ and $3\times$ Mask R-CNN (He et al., 2017) training settings in Swin Transformer (Liu et al., 2021) to conduct our experiments. The pretrained ViF models on ImageNet-1K are employed as backbones and fine-tuned on COCO. For the $1\times$ schedule, we train for 12 epochs with an initial learning rate of $2\times10^{-4}$ and decay the learning rate by 10 at epochs 8 and 11. For the $3\times$ schedule, we extend the training to 36 epochs with learning rate decay at epochs 27 and 33. We use AdamW optimizer with a weight decay of 0.05 and a batch size of 16. Data augmentation techniques including horizontal flipping are employed during training.

**ADE20K for semantic segmentation.** The ADE20K dataset (Zhou et al., 2019) comprises 20K images for training, 2K images for validation, and 3K images for testing, with 150 semantic categories. We follow the standard UPerNet (Xiao et al., 2018) training settings in Swin Transformer (Liu et al., 2021) to conduct our experiments. The pretrained ViF models on ImageNet-1K are employed as backbones and fine-tuned on ADE20K. We train for 160K iterations with an initial learning rate of $6 \times 10^{-5}$ using AdamW optimizer. The learning rate follows a polynomial decay schedule with a power of 0.9. We set the weight decay to $1 \times 10^{-4}$ and use a batch size of 16. Data augmentation includes random horizontal flipping, random resizing with scale range [0.5, 2.0], and random cropping

to 512×512 pixels. We report both single-scale and multi-scale testing results, where multi-scale testing uses scales of [0.5, 0.75, 1.0, 1.25, 1.5, 1.75] with horizontal flipping.

# D    ARCHITECTURE DETAILS

The detailed architectures of ViF models are outlined in Tab. A1. Following the common four-stage hierarchical framework Liu et al. (2021), we construct the ViF models by stacking our proposed ViF blocks at each stage. Specifically, an input image with resolution of $224 \times 224$ is firstly processed by a stem layer, which consists of Convolution (Conv), Batch Normalization (BN) and GELU activation function. The kernel size is $3 \times 3$ with a stride of 2 at the first and last convolution layers, and a stride of 1 for other layers. Each stage contains multiple ViF blocks, followed by a down-sampling layer except for the last block. The down-sampling layer consists of a $3 \times 3$ convolution with a stride of 2 and a Layer Normalization (LN) layer. Each block incorporates a FNF module and a FFN module, both with residual connections. The FNF module has two branches: one branch contains a local convolution and a global convolution, the other branch contains a local convolution, where the local convolution is a depth-wise convolution layer with the kernel of $3 \times 3$. The expand ratio of SSM is set to 2, doubling the number of channels. We modify the embedding dimension and number of blocks to build our ViF-T/S/B models.

Table A1: **Architecture details of our ViF models.**

| Layer | Output size | ViF-T | ViF-S | ViF-B |
|---|---|---|---|---|
| Stem | $56 \times 56$ | Conv $3 \times 3$ stride 2, BN, GELU; Conv $3 \times 3$ stride 1, BN; Conv $3 \times 3$ stride 2, BN | | |
| Stage1 | $28 \times 28$ | ViF Blocks $\begin{bmatrix} \text{Linear } 64 \to 128 \\ \text{LocalConv } 128 \\ \text{GlobalConv } 128 \\ \text{Linear } 128 \to 64 \\ \text{FFN } 64 \end{bmatrix} \times 2$ | ViF Blocks $\begin{bmatrix} \text{Linear } 64 \to 128 \\ \text{LocalConv } 128 \\ \text{GlobalConv } 128 \\ \text{Linear } 128 \to 64 \\ \text{FFN } 64 \end{bmatrix} \times 2$ | ViF Blocks $\begin{bmatrix} \text{Linear } 96 \to 192 \\ \text{LocalConv } 192 \\ \text{GlobalConv } 192 \\ \text{Linear } 192 \to 96 \\ \text{FFN } 96 \end{bmatrix} \times 2$ |
| | | Down Sampling Conv $3 \times 3$ stride 2, LN | | |
| Stage2 | $14 \times 14$ | ViF Blocks $\begin{bmatrix} \text{Linear } 128 \to 256 \\ \text{LocalConv } 256 \\ \text{GlobalConv } 256 \\ \text{Linear } 256 \to 128 \\ \text{FFN } 128 \end{bmatrix} \times 4$ | ViF Blocks $\begin{bmatrix} \text{Linear } 128 \to 256 \\ \text{LocalConv } 256 \\ \text{GlobalConv } 256 \\ \text{Linear } 256 \to 128 \\ \text{FFN } 128 \end{bmatrix} \times 5$ | ViF Blocks $\begin{bmatrix} \text{Linear } 192 \to 384 \\ \text{LocalConv } 384 \\ \text{GlobalConv } 384 \\ \text{Linear } 384 \to 192 \\ \text{FFN } 192 \end{bmatrix} \times 5$ |
| | | Down Sampling Conv $3 \times 3$ stride 2, LN | | |
| Stage3 | $7 \times 7$ | ViF Blocks $\begin{bmatrix} \text{Linear } 256 \to 512 \\ \text{LocalConv } 512 \\ \text{GlobalConv } 512 \\ \text{Linear } 512 \to 256 \\ \text{FFN } 256 \end{bmatrix} \times 8$ | ViF Blocks $\begin{bmatrix} \text{Linear } 256 \to 512 \\ \text{LocalConv } 512 \\ \text{GlobalConv } 512 \\ \text{Linear } 512 \to 256 \\ \text{FFN } 256 \end{bmatrix} \times 19$ | ViF Blocks $\begin{bmatrix} \text{Linear } 384 \to 768 \\ \text{LocalConv } 768 \\ \text{GlobalConv } 768 \\ \text{Linear } 768 \to 384 \\ \text{FFN } 384 \end{bmatrix} \times 19$ |
| | | Down Sampling Conv $3 \times 3$ stride 2, LN | | |
| Stage4 | $7 \times 7$ | ViF Blocks $\begin{bmatrix} \text{Linear } 512 \to 1024 \\ \text{LocalConv } 1024 \\ \text{GlobalConv } 1024 \\ \text{Linear } 1024 \to 512 \\ \text{FFN } 512 \end{bmatrix} \times 4$ | ViF Blocks $\begin{bmatrix} \text{Linear } 512 \to 1024 \\ \text{LocalConv } 1024 \\ \text{GlobalConv } 1024 \\ \text{Linear } 1024 \to 512 \\ \text{FFN } 512 \end{bmatrix} \times 5$ | ViF Blocks $\begin{bmatrix} \text{Linear } 768 \to 1536 \\ \text{LocalConv } 1536 \\ \text{GlobalConv } 1536 \\ \text{Linear } 1536 \to 768 \\ \text{FFN } 768 \end{bmatrix} \times 5$ |
| Head | $1 \times 1$ | Average pool, Linear 1000, Softmax | | |

# E    SPATIAL AND FREQUENCY ANALYSIS

To quantitatively assess the influence of different modules (SA and AM) on feature representations, we perform comprehensive spatial- and frequency-domain analyses on the fourth-layer features of our model.

## E.1    SPATIAL CORRELATION ANALYSIS

Given a feature map $\mathbf{F} \in \mathbb{R}^{C \times H \times W}$ from the fourth layer, we compute spatial correlation to characterize structural relationships across spatial locations.

We first reshape the feature map into a matrix $\mathbf{X} \in \mathbb{R}^{(H \cdot W) \times C}$, where each row corresponds to the feature vector at a specific spatial position. Each vector is then normalized as:

$$\mathbf{X}_{\text{norm}}(i) = \frac{\mathbf{X}(i)}{\|\mathbf{X}(i)\|_2 + \epsilon}, \tag{A1}$$

where $i \in \{1, 2, \ldots, H \cdot W\}$ indexes spatial locations and $\epsilon = 10^{-6}$ prevents division by zero.

The spatial correlation matrix $\mathbf{R} \in \mathbb{R}^{(H \cdot W) \times (H \cdot W)}$ is computed by:

$$\mathbf{R} = \mathbf{X}_{\text{norm}} \mathbf{X}_{\text{norm}}^T, \tag{A2}$$

where $\mathbf{R}(i, j)$ measures the correlation between spatial positions $i$ and $j$.

To obtain a per-pixel correlation response, we average the correlation of each location with all others:

$$\mathcal{C}(i) = \frac{1}{H \cdot W} \sum_{j=1}^{H \cdot W} \mathbf{R}(i, j). \tag{A3}$$

We then reshape $\mathcal{C}$ to spatial dimensions and normalize it:

$$\mathcal{C}_{\text{map}} = \text{Reshape}(\mathcal{C}, H, W), \quad \mathcal{C}_{\text{norm}} = \frac{\mathcal{C}_{\text{map}} - \min(\mathcal{C}_{\text{map}})}{\max(\mathcal{C}_{\text{map}}) - \min(\mathcal{C}_{\text{map}})}. \tag{A4}$$

Higher values in $\mathcal{C}_{\text{norm}}$ indicate spatial locations that exhibit stronger global correlation, reflecting the model's capacity to capture structural and contextual dependencies.

**Structural Difference Metric.** To quantify structural differences between models, we compute the Pearson correlation coefficient between correlation maps:

$$\Delta_{\text{struct}} = (1 - \rho(\mathcal{C}_{\text{full}}, \mathcal{C}_{\text{ablation}})) \times 100\%, \tag{A5}$$

where $\rho$ denotes the Pearson correlation coefficient. Larger $\Delta_{\text{struct}}$ indicates greater structural discrepancy.

### E.2 FREQUENCY SPECTRUM ANALYSIS

To examine the frequency characteristics of learned representations, we apply a Fourier transform to the activation maps. For a given feature map $\mathbf{F} \in \mathbb{R}^{C \times H \times W}$, we first compute the spatial activation:

$$\mathbf{A} = \frac{1}{C} \sum_{c=1}^{C} \mathbf{F}_c^2. \tag{A6}$$

We then apply a 2D Fourier transform and compute the magnitude spectrum:

$$\mathcal{F} = \text{FFTShift}(\text{FFT2D}(\mathbf{A})), \tag{A7}$$

$$\mathcal{M} = 20 \cdot \log(|\mathcal{F}| + 1), \tag{A8}$$

where FFTShift centers the zero-frequency component and logarithmic scaling improves visual interpretability.

**Frequency Energy Distribution.** The spectrum is decomposed into three frequency bands according to the radial distance $r$ from the center:

$$E_{\text{low}} = \sum_{r < 0.1 r_{\text{max}}} \mathcal{M}(r), \tag{A9}$$

$$E_{\text{mid}} = \sum_{0.1 r_{\text{max}} \leqslant r \leqslant 0.3 r_{\text{max}}} \mathcal{M}(r), \tag{A10}$$

$$E_{\text{high}} = \sum_{r > 0.3 r_{\text{max}}} \mathcal{M}(r), \tag{A11}$$

where $r_{\max} = \min(H/2, W/2)$ denotes the maximum radius.

The frequency shift metric is defined as:

$$\Delta_{\text{freq}} = \frac{(E_{\text{mid}}^{\text{ablation}} + E_{\text{high}}^{\text{ablation}}) - (E_{\text{mid}}^{\text{full}} + E_{\text{high}}^{\text{full}})}{E_{\text{total}}^{\text{full}}} \times 100\%. \tag{A12}$$

A positive $\Delta_{\text{freq}}$ indicates that the ablated model retains more high-frequency components, whereas negative values suggest that the full model better preserves fine-grained details.

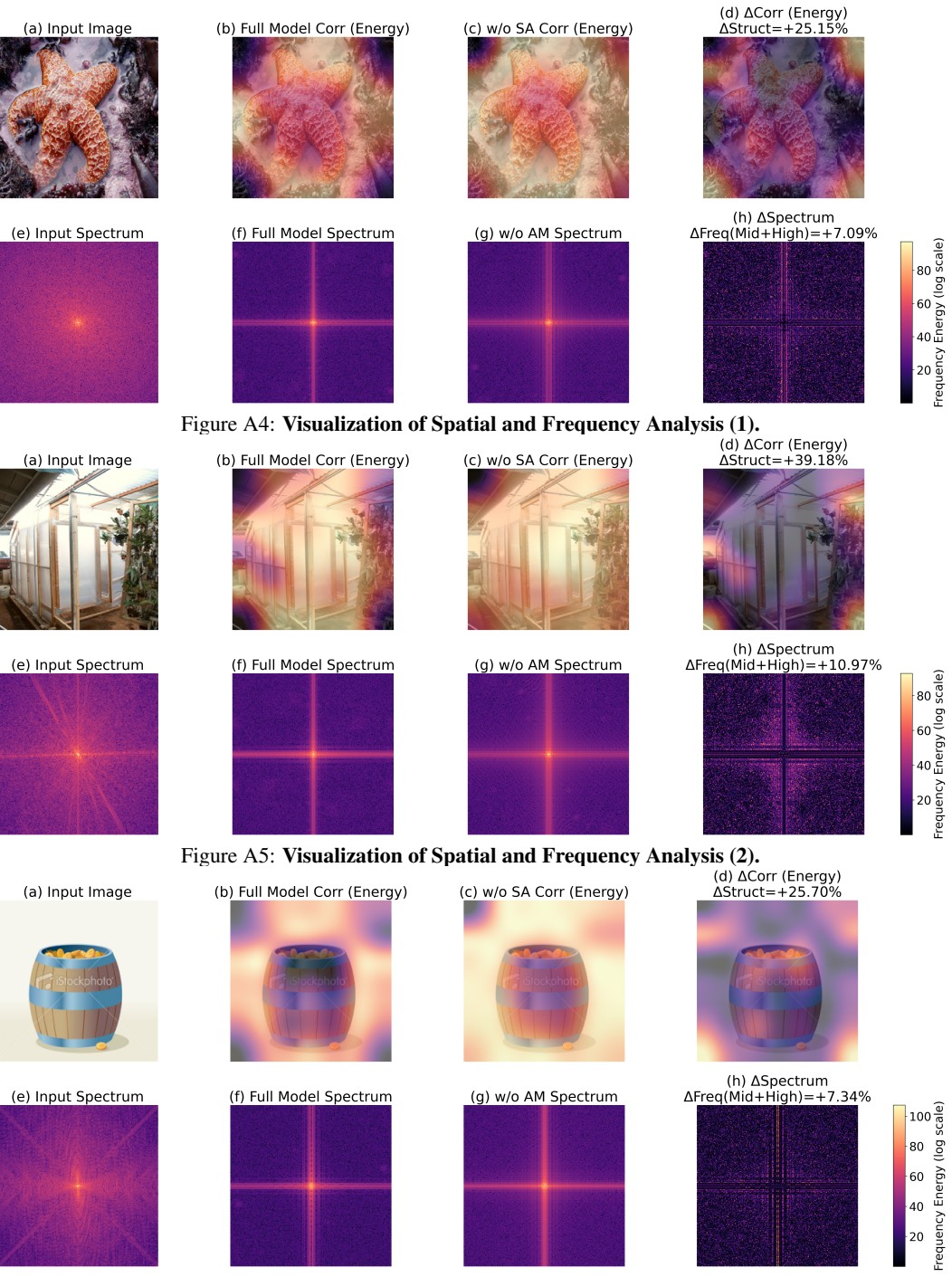

Figure A4: **Visualization of Spatial and Frequency Analysis (1).**

Figure A5: **Visualization of Spatial and Frequency Analysis (2).**

Figure A6: **Visualization of Spatial and Frequency Analysis (3).**

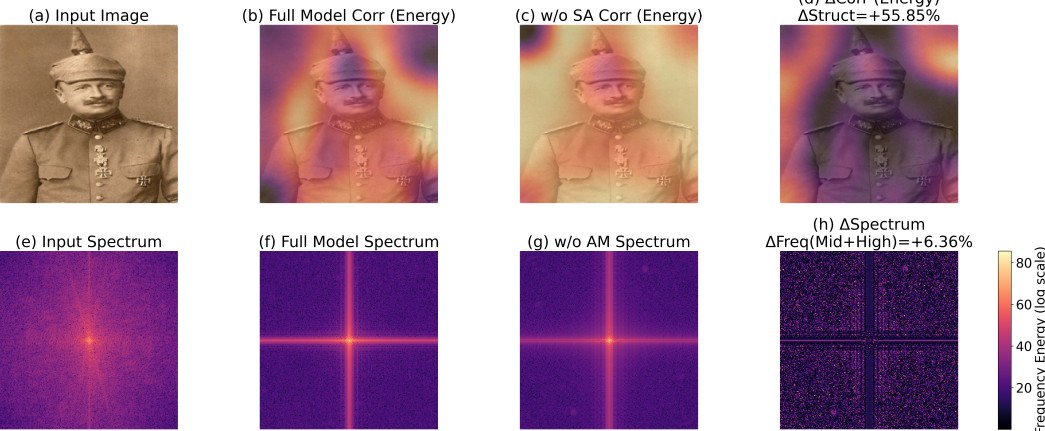

Figure A7: **Visualization of Spatial and Frequency Analysis (4).**

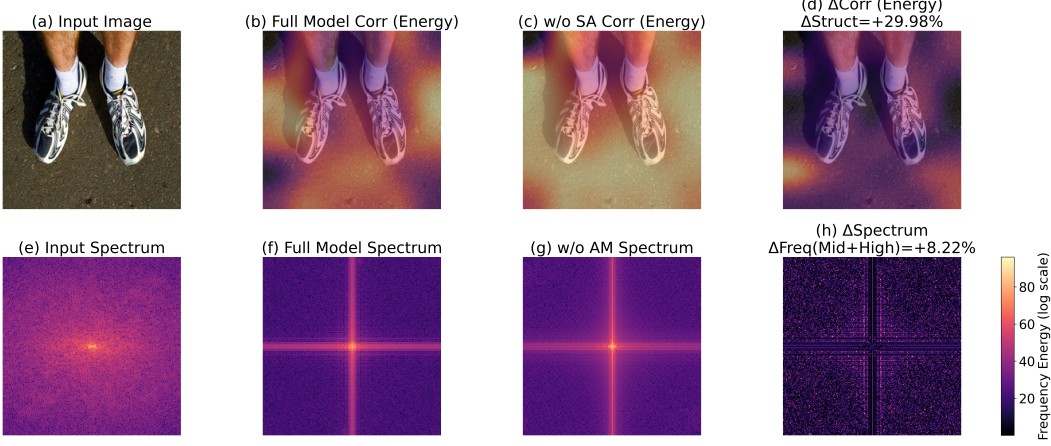

Figure A8: **Visualization of Spatial and Frequency Analysis (5).**

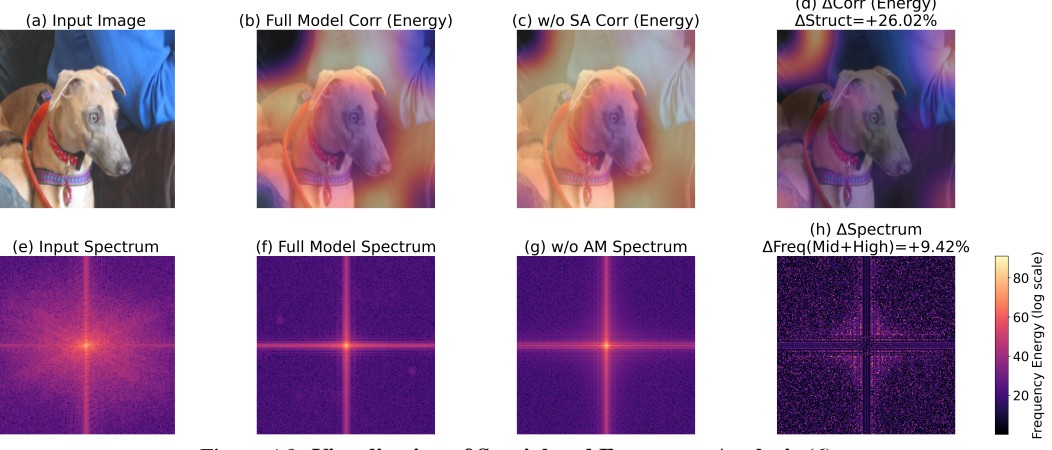

Figure A9: **Visualization of Spatial and Frequency Analysis (6).**

# F    EFFECTIVE RECEPTIVE FIELD COMPARISON

We compare the effective receptive field (ERF) characteristics of three representative architectures: Transformer, Fourier Neural Field (FNF), and Mamba with different scanning strategies (Figure X). Transformer employs self-attention mechanisms that establish all-to-all connections between tokens, achieving a global receptive field from the first layer. However, this comes at the cost of $O(N^2)$ computational complexity, where N is the sequence length, making it computationally prohibitive for long sequences exceeding several thousand tokens. FNF leverages Fourier transforms to operate in the frequency domain, enabling global receptive field coverage through spectral convolutions. By transforming inputs via Fast Fourier Transform (FFT), mixing in frequency space, and applying inverse FFT, FNF achieves global modeling with $O(N \log N)$ complexity. This approach provides an efficient middle ground between local convolutions and quadratic attention mechanisms.

Mamba employs selective state space models (SSMs) with linear $O(N)$ complexity, making it highly efficient for long sequences. Unlike Transformer and FNF, Mamba's receptive field is not immediately global from the first layer but rather develops through recurrent state propagation. We visualize three scanning variants: (1) Unidirectional scanning follows a causal dependency pattern, suitable for autoregressive tasks; (2) Bidirectional scanning enables two-way information flow, enhancing contextual understanding; (3) Quad-directional scanning performs multi-way traversal (top-down, bottom-up, left-right, right-left), approximating more uniform spatial coverage comparable to global attention.

As shown in our complexity analysis, Mamba maintains constant per-token cost regardless of sequence length, while Transformer's cost grows quadratically and FNF's grows quasi-linearly. The ERF visualizations demonstrate that while Transformer and FNF achieve uniform global coverage, Mamba variants exhibit distance-dependent patterns influenced by their scanning strategies, with quad-directional scanning providing the most balanced spatial coverage among Mamba variants.

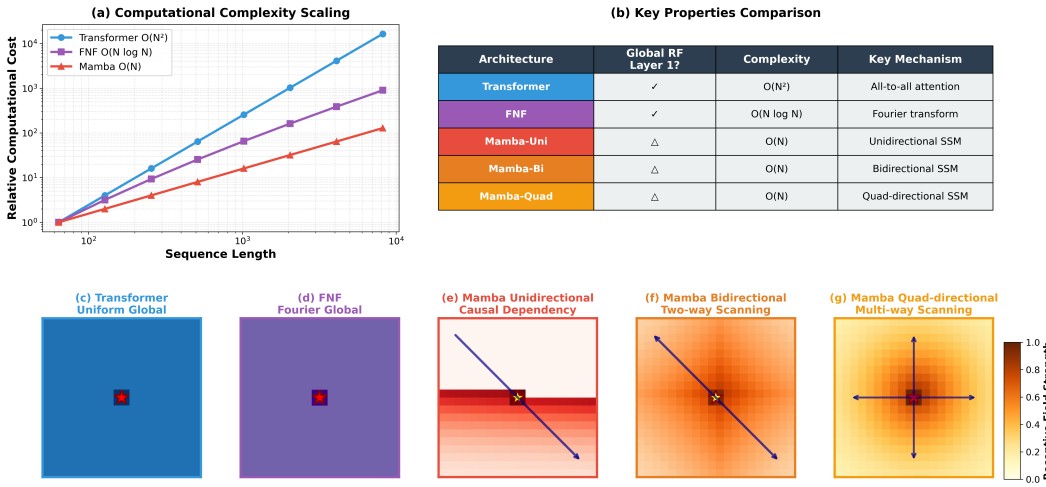

Figure A10: **Effective Receptive Field Comparison.**