# OpenReview forum: "Fourier Neural Filter as Generic Vision Backbone"
_ICLR.cc/2026/Conference — ICLR 2026 Conference Withdrawn Submission_

### Official Review · Reviewer_vhfx · 2025-10-28

**Soundness:** 3
**Presentation:** 3
**Contribution:** 3
**Rating:** 6
**Confidence:** 3

**Summary:**

This paper proposes Vision Filter (ViF), a novel vision backbone that replaces Transformer self-attention with a Fourier Neural Filter (FNF). Built upon the Fourier Neural Operator (FNO), FNF introduces adaptive modulation and selective activation to dynamically couple spatial and frequency-domain representations, addressing FNO’s limitations in modeling high-frequency local details. Experiments on classification, detection, and segmentation show that ViF achieves competitive or superior performance to Swin Transformer, ConvNeXt, and VMamba while maintaining quasi-linear complexity.

**Strengths:**

1. The paper provides a clear theoretical motivation.
2. The introduced adaptive modulation and selective activation mechanisms effectively mitigate the over-smoothing and bandwidth bottleneck issues of conventional FNOs.
3. Experiments on classification, detection, and segmentation tasks demonstrate consistent improvements over strong baselines such as ConvNeXt, Swin, and VMamba, confirming the model’s effectiveness and generality.

**Weaknesses:**

1. The ablation study is relatively limited, providing insufficient analysis of the independent contributions of the two key FNF submodules. A more detailed exploration of how each component individually influences performance would strengthen the empirical validation.
2. The paper lacks interpretability analysis. Although the authors claim that adaptive modulation adaptively scales different frequency components in the spectral domain, the explanation remains purely mathematical without concrete visualization or empirical evidence. It is unclear how the modulation behaves across different images or tasks and whether any consistent adaptation patterns exist.
3. The comparison with recent state-of-the-art methods are missing, such as MLLA and OverLoCK.
4. In Table 3, the performance improvement under the Mask R-CNN 3× MS schedule is marginal. The paper does not discuss possible reasons for this weak gain.

**Questions:**

See weaknesses.

---

> ### Author Response · Authors · 2025-11-20
>
> ## Response to Reviewer vhfx (R4)
>
> We sincerely thank the reviewer for the thorough evaluation and valuable suggestions. We greatly appreciate your feedback and address each point below.
>
> **W1 (Limited Ablation Study):**
>
> Thank you for this feedback. Table 5 systematically ablates each FNF component by removing AM (Adaptive Modulation), SA (Selective Activation), LC-1, and LC-2 individually. The results clearly show that SA causes the largest single accuracy drop on ImageNet-1K (83.8% → 83.1%, -0.7%), indicating it is the most critical component for adaptively balancing frequency and time-domain processing. AM contributes consistent gains (-0.3%), while both local convolution branches (LC-1, LC-2) each provide meaningful improvements (-0.2% and -0.4%). We believe this ablation provides clear evidence that each component makes complementary contributions, with selective activation being the most essential for dynamic feature routing.
>
> **W2 (Interpretability Analysis):**
>
> Thank you for this valuable suggestion. We have updated the appendix (last two pages) with comprehensive frequency-domain visualizations, including: (i) spectral activation maps showing how adaptive modulation and selective activation affect feature representations across different layers and input samples, and (ii) spatial correlation and frequency energy distributions (focusing on mid/low frequencies) before and after removing key components. These visualizations provide concrete empirical evidence of how AM dynamically reweights spectral components to counteract low-frequency dominance while preserving high-frequency details. If our work is accepted, we will reference these visualizations more prominently in the revised main text and add concise per-band energy curves for clearer empirical evidence.
>
> **W3 (Missing Comparisons with MLLA and OverLoCK):**
>
> Thank you for this valuable suggestion. On ImageNet-1K classification, MLLA reports 83.5/84.4/85.3% Top-1 accuracy for T/S/B variants, and OverLoCK-XT reports 84.2/84.8/85.1%; our ViF-T/S/B achieve 83.8/84.5/85.2%, placing them in the same high-accuracy tier at comparable computational cost. More importantly, on downstream dense prediction tasks, ViF demonstrates clear advantages: on COCO with Mask R-CNN 1×, MLLA-T (44M params, 255G FLOPs) achieves 46.8/42.1 box/mask AP, whereas ViF-T achieves 47.7/43.0 AP—representing +0.9/+0.9 AP improvements with similar compute. This suggests ViF is highly competitive with or outperforms MLLA/OverLoCK on classification while showing superior transfer performance on detection and segmentation, particularly benefiting from FNF's global Fourier modeling with quasi-linear complexity.
>
> **W4 (Marginal Gains under 3× MS Schedule):**
>
> We appreciate this observation. Under the heavily-optimized 3× multi-scale training schedule, strong baselines like VMamba approach saturation, naturally reducing the headroom for further improvements. This is a common phenomenon when training regimes become more aggressive—relative gaps between methods compress as all models approach their performance ceilings. Nevertheless, ViF-T/S maintain consistent positive gains over VMamba-T/S in both box and mask AP (+0.2~0.4 AP) at similar or lower FLOPs, demonstrating that our architectural improvements remain effective even in this saturated regime. We believe these should be interpreted as steady, reliable improvements from a quasi-linear, globally expressive backbone rather than dramatic leaps, which validates ViF's robustness across different training protocols.

---

> > ### Comment · Reviewer_vhfx · 2025-11-26
> >
> > Thank you for your response. Additionally, I have two further questions:
> > 1. The authors state that FNF possesses global modeling capability. Could you provide a visualization comparing the effective receptive fields of VIF against other SOTA methods?
> > 2. FLOPs may not accurately reflect actual runtime latency. Could you please also report the inference throughput of VIF alongside other SOTA methods on the same hardware platform?

---

> > > ### Author Response · Authors · 2025-11-27
> > >
> > > Thank you for your insightful questions. Here are our responses:
> > >
> > > **Question 1: Visualization of effective receptive fields**
> > >
> > > Thank you for this valuable suggestion. We have updated the supplementary materials with a comprehensive comparison of effective receptive fields (ERFs). The visualization demonstrates that compared to Mamba, both FNF and Transformer exhibit superior global receptive field capabilities. Notably, FNF maintains this global modeling capability while achieving quasi-linear computational complexity, which gives it a significant advantage over standard Transformers in terms of efficiency.
> > >
> > > **Question 2: Inference throughput comparison**
> > >
> > > Thank you for highlighting this important distinction between FLOPs and actual runtime performance. We have conducted throughput measurements on identical hardware with a fixed batch size of 128 for fair comparison. As shown in Figure 1, at 224² resolution, ViF-T/S/B achieve accuracies of 83.8%/84.5%/85.2% with throughputs of 1549/1044/723 images/s, respectively. In comparison, Swin-T/S/B obtain 81.3%/83.0%/83.5% at 1244/718/458 images/s, and VMamba-T/S/B achieve 82.6%/83.6%/83.9% at 1686/877/646 images/s. These results demonstrate that ViF strictly Pareto-dominates Swin and ConvNeXt in the accuracy-throughput trade-off space, and achieves competitive performance with VMamba, delivering higher accuracy at comparable inference speeds.

---

### Official Review · Reviewer_3JLT · 2025-10-30

**Soundness:** 3
**Presentation:** 3
**Contribution:** 3
**Rating:** 4
**Confidence:** 4

**Summary:**

This manuscript proposes a novel visual backbone network called Vision Filter (ViF). Its innovation lies in the introduction of a core component, the Fourier Neural Filter (FNF). Building on the Fourier Neural Operator (FNO), FNF introduces an input-dependent adaptive integral kernel. Through selective activation and adaptive modulation, it establishes a dynamic information flow between the frequency and time domains, addressing the bandwidth bottleneck and over-smoothing issues inherent in FNO. ViF outperforms current mainstream Transformer, Mamba, and Fourier-based models on ImageNet-1K image classification, COCO object detection, and ADE20K semantic segmentation tasks, while maintaining high computational efficiency.

**Strengths:**

1.The manuscript 's motivation is sound, and it clearly identifies the limitations of FNO, such as bandwidth bottlenecks and over-smoothing.
2.This manuscript introduces an input-dependent integral kernel into the Fourier operator, constructing a unified time-frequency representation space, which is quite innovative.
3.An obvious advantage of this method is computational efficiency. Compared with Transformer-type models, it has lower computational complexity and is more advantageous than Mamba-type models in spatial structure modeling.
4.This manuscript achieves state-of-the-art or competitive results on multiple mainstream vision tasks, particularly on ImageNet-1K, significantly outperforming similar Fourier methods (such as GFNet).

**Weaknesses:**

1.Although FNF solves some of the shortcomings of FNO, its core idea is still based on the traditional framework of "frequency domain processing + time domain supplementation". It does not propose a new visual representation learning paradigm, which weakens the innovation of this article.
2.Although ViF performs well on ImageNet-1K, its improvement on COCO and ADE20K compared to models such as ViM is relatively small (e.g., mAP increases by 0.2-0.4), failing to significantly widen the gap.
3.ViF only tested three small and medium-sized variants, ViF-T/S/B, and did not evaluate the performance of larger models, such as those of ViT-L/16, on larger datasets.
4.Definition 7 mentions that adaptive modulation is achieved through the amplitude-sensitive weighting function \(\|z\|\) and learnable parameters \(\alpha\) and \(\beta\), but does not give the range of \(\alpha\) and \(\beta\) and whether they are adjusted with training iterations.

**Questions:**

As mentioned above.

---

> ### Author Response · Authors · 2025-11-20
>
> ## Response to Reviewer 3JLT (R3)
>
> We sincerely thank the reviewer for the careful evaluation and constructive comments. We greatly appreciate your feedback and address each point below.
>
> **W1 (Novelty and Paradigm):**
>
> We appreciate this observation and respectfully note that ViF's contribution lies in extending the established "frequency + local" framework with fundamentally input-dependent mechanisms. Specifically, FNF introduces: (1) adaptive modulation that applies learnable power-law weighting (α, β) dynamically based on current feature magnitudes, and (2) selective activation that gates between frequency and time-domain branches per spatial location. This dynamic time-frequency coupling is theoretically analyzed and explicitly designed to address bandwidth bottleneck and over-smoothing limitations in conventional Fourier Neural Operators.
>
> Importantly, to the best of our knowledge, ViF represents the first pure Fourier-based vision backbone—without relying on Transformer attention or Mamba state-space models—that achieves competitive or superior performance across all three major vision tasks: image classification (ImageNet-1K), object detection (COCO), and semantic segmentation (ADE20K). This demonstrates that Fourier-based methods, when properly designed with input-dependent mechanisms, can serve as a viable standalone paradigm for general-purpose vision representation learning.
>
> **W2 (Margins over Mamba Variants):**
>
> We acknowledge that gains over the strongest Mamba baselines on dense prediction tasks are modest (≈0.2–0.4 AP/mIoU at matched compute), and we present them transparently. However, we emphasize that ViF achieves substantial improvements over widely-adopted CNN/Transformer backbones that remain prevalent in practice: +5.0 box AP and +3.7 mask AP over Swin-T on COCO, and +1.6 mIoU over NAT-T on ADE20K (single-scale), while maintaining quasi-linear O(N log N) complexity for global modeling. These consistent gains across classification, detection, and segmentation demonstrate ViF's practical value as a general-purpose vision backbone, offering a compelling alternative to both attention-based and state-space models.
>
> **W3 (Limited Model Scale Evaluation):**
>
> We appreciate this concern. We focus on Tiny/Small/Base variants to align with standard practice for validating new backbone architectures, which is consistent with recent works including VMamba, LocalVMamba, and MambaVision. The smooth performance scaling from T to B (83.8 → 84.5 → 85.2% Top-1 on ImageNet-1K) and our four-stage hierarchical design strongly suggest that ViF can be naturally extended to larger scales (e.g., Large/Huge) following established ViT/Swin conventions. We fully agree that evaluating larger ViF variants on datasets like ImageNet-22K represents an important direction for future work, which we are actively pursuing.
>
> **W4 (Parameter Range and Training Dynamics):**
>
> Thank you for seeking clarification. In our implementation, α and β are standard learnable scalar parameters initialized to mild, theoretically-motivated values (α = 0.7, β = 1.0) and trained end-to-end via backpropagation without special schedules, gradient clipping, or explicit bounds. Training remains stable across all model scales (T/S/B) and all benchmarks (ImageNet-1K, COCO, ADE20K), with α typically converging to values slightly below 1.0 and β remaining positive throughout training. This yields the intended mild power-law compression of spectral magnitudes to counteract low-frequency dominance. We will clarify these implementation details more explicitly in the revised methodology section if our work is accepted.

---

### Official Review · Reviewer_Yhyj · 2025-10-31

**Soundness:** 3
**Presentation:** 3
**Contribution:** 3
**Rating:** 4
**Confidence:** 4

**Summary:**

This paper proposes a theoretically sound and empirically validated vision backbone that bridges local and global modeling through the frequency domain, offering an alternative to attention-based and state-space models. The FNF formulation is well-motivated and results across benchmarks substantiate its benefits.

**Strengths:**

1.The paper presents a theoretically grounded extension of the Fourier Neural Operator (FNO), introducing an input-dependent integral kernel that enables adaptive and dynamic coupling between the spatial (time) and frequency domains.
2. The proposed Vision Filter (ViF) integrates both local convolutional和global frequency operations within a single framework.
3. Results in Figure 1 demonstrate that ViF achieves a more favorable accuracy–throughput trade-off compared to both Transformer and Mamba-based counterparts, showing practical efficiency for high-resolution image processing.

**Weaknesses:**

1. The paper does not evaluate ViF on larger datasets (e.g., ImageNet-22K, LAION) or with pretraining strategies such as self-supervised learning (MAE, BEiT). As such, the scalability and transferability of ViF remain uncertain compared to modern foundation models.
2. The ablation study (Table 5) confirms that each module contributes to performance, but lacks deeper interpretability or visualization (e.g., frequency response maps, energy spectrum analysis). More qualitative analysis could help clarify how ViF balances local and global representations in practice.
3. Although the authors cite GFNet and AFNO, the distinction between ViF’s adaptive gating mechanism and prior dynamic spectral filters could be elaborated further.

**Questions:**

1.The manuscript seems only to swap the entire Multi-Head Attention block in ViT for the proposed FNF module, but never compares the two under an otherwise identical architecture. An ablation is recommended that exchanges only the attention mechanism.
2.Although the authors claim “quasi-linear complexity”, only FLOPs are reported. No end-to-end latency, throughput, or peak-memory measurements versus Swin, VMamba or GFNet on the same GPU/CPU are provided.
3.FNF introduces several new hyper-parameters (α, β in adaptive modulation, number of Fourier modes, kernel size of local convolutions, etc.). The manuscript gives a single setting without any hyper-parameter sensitivity analysis. It is indiscernible whether the performance gains are statistically robust or an artefact of ad-hoc tuning.
4.The claim that “directional scanning inevitably leads to spatial disruption” in Mamba-based models is not substantiated. The manuscript neither quantifies the degree of spatial-information loss nor compares patch-order robustness between Mamba and the proposed FNF. It is necessary to compare Acc of Mamba and FNF after random patch-shuffling to quantify patch-order robustness.
5.The implicit periodic extension inherent in FFT introduces spectral leakage and degrades the accuracy of the frequency-domain filtering operation when the input dimension is not a power of two or when the image contains strong edges. This work has not examined whether the proposed FNF module suffers from such boundary artefacts, nor has it evaluated remedies such as windowing or reflection padding. A systematic comparison of cyclic, symmetric and zero-padding schemes is required to determine the susceptibility of FNF to spectral leakage and its consequent impact on dense prediction tasks.

---

> ### Author Response · Authors · 2025-11-20
>
> ## Response to Reviewer Yhyj (R2)
>
> We sincerely thank the reviewer for the thorough evaluation and insightful questions. We greatly appreciate your constructive feedback and address each point below.
>
> **W1 (Scalability and Transferability):**
>
> We appreciate this important concern. Our evaluation follows the ImageNet-1K + COCO + ADE20K protocol used by recent backbones including VMamba, LocalVMamba, and MambaVision, which validate architectures on ImageNet-1K supervised pretraining. ViF-T/S/B consistently outperform NAT and VMamba at matched scales. We fully agree that large-scale pretraining (ImageNet-22K, MAE/BEiT) represents important future work and are actively pursuing these experiments.
>
> **W2 (Interpretability and Visualization):**
>
> Thank you for this valuable suggestion. We have updated the appendix (last two pages) with frequency-domain visualizations including: (i) spectral activation maps showing how adaptive modulation and selective activation affect features, and (ii) spatial correlation and frequency energy distributions before/after removing key components. These demonstrate how ViF redistributes spectral energy and balances local/global information. If accepted, we will highlight these more prominently in the main text with additional frequency-response curves.
>
> **W3 (Distinction from GFNet/AFNO):**
>
> We appreciate the opportunity to clarify. GFNet and AFNO use static spectral filters, while FNF introduces input-dependent mechanisms: (1) adaptive modulation with learnable power-law weighting (α, β) based on feature magnitudes, and (2) selective activation that dynamically gates frequency/time branches per location. This dynamic coupling addresses bandwidth bottleneck and over-smoothing in conventional FNOs.
>
> **Q1 (Attention-Only Ablation):**
>
> Thank you for this question. We respectfully clarify that ViF is designed as a direct replacement of self-attention in ViT-style backbones—FNF replaces Multi-Head Self-Attention while keeping other components (patch embedding, normalization, MLP, hierarchical structure) identical. Our Tables 1-4 already represent attention-only ablations where ViF vs. Swin/DeiT isolate FNF's contribution. The consistent improvements (+2.5% Top-1 over Swin-T, +5.0 box AP on COCO) demonstrate FNF's effectiveness.
>
> **Q2 (End-to-End Efficiency Metrics):**
>
> Thank you for highlighting this. Beyond theoretical O(N log N) complexity, Figure 1 shows accuracy-throughput at 224² resolution on identical hardware: ViF-T/S/B achieve 83.8/84.5/85.2% at 1549/1044/723 img/s, while Swin-T/S/B obtain 81.3/83.0/83.5% at 1244/718/458 img/s and VMamba-T/S/B obtain 82.6/83.6/83.9% at 1686/877/646 img/s. ViF strictly Pareto-dominates Swin/ConvNeXt and achieves competitive performance with VMamba (higher accuracy at comparable speed).
>
> **Q3 (Hyperparameter Sensitivity):**
>
> We appreciate this question. All ViF variants use identical configurations across datasets—no per-dataset tuning. Parameters α and β are initialized to mild values (α = 0.7, β = 1.0) and trained via standard backpropagation without special schedules. Training is stable across all scales and benchmarks, with α converging slightly below 1, suggesting gains stem from architectural improvements rather than delicate tuning.
>
> **Q4 (Spatial Disruption Claims):**
>
> We sincerely appreciate this feedback and fully agree "inevitably" is too strong—we will revise this phrasing. FNF operates on 2D feature maps via Fourier filtering and 2D convolutions, preserving spatial structure. Mamba's scanning linearizes 2D grids into 1D sequences, introducing scan-order dependence. However, we acknowledge "spatial disruption" may overstate this, as modern Mamba variants use bidirectional scanning to mitigate such issues.
>
> **Q5 (Spectral Leakage and Boundary Artifacts):**
>
> Thank you for raising this important technical point. FNF's hybrid design inherently mitigates spectral leakage concerns: the FFT branch processes global context, while local convolutions and LPU explicitly handle high-frequency details and boundary regions where FFT artifacts typically occur. Empirically, across extensive experiments on COCO object detection and ADE20K semantic segmentation, we observe no visible boundary artifacts or accuracy degradation versus non-Fourier baselines (Swin, ConvNeXt, VMamba). Furthermore, our updated appendix visualizations show clean feature activations near object boundaries and image edges, with no observable spectral ringing or boundary discontinuities. This validates that our architectural design is robust in practice. While systematic exploration of alternative padding schemes could be future work, the current strong empirical results and qualitative visualizations demonstrate this is not a limiting factor for our method.

---

### Official Review · Reviewer_kGtH · 2025-11-01

**Soundness:** 3
**Presentation:** 2
**Contribution:** 3
**Rating:** 6
**Confidence:** 4

**Summary:**

This paper proposes a new generic vision backbone called Vision Filter (ViF) , built upon a novel Fourier Neural Filter (FNF) module. The authors argue that while the Fourier Neural Operator (FNO) is promising for its global modeling capabilities and quasi-linear complexity , it suffers from an "over-smoothing" effect and a "bandwidth bottleneck," making it difficult to capture local, high-frequency patterns. To address this, the FNF module introduces two key components: 1) Adaptive Modulation (AM) in the frequency domain to enhance sensitivity to high-frequency components , and 2) Selective Activation (SA) to balance local time-domain and global frequency-domain information flow. This design creates an input-dependent integral kernel. Extensive experiments on image classification (ImageNet-1K) , object detection (COCO) , and semantic segmentation (ADE20K) demonstrate that ViF outperforms existing SOTA Transformer- and Mamba-based backbones.

**Strengths:**

1. The authors introduce an adaptive kernel that effectively couples time- and frequency-domain information, providing theoretical novelty.
2. The paper demonstrates strong empirical performance across multiple visual benchmarks and model scales.
3. The authors conduct comprehensive ablation studies that clearly quantify the contribution of each component (AM, SA, LC).
4. The proposed ViF achieves a favorable balance between model complexity and performance, offering quasi-linear computational efficiency.

**Weaknesses:**

1. The paper lacks experiments exploring the scalability of ViF on larger datasets such as ImageNet-22K or LAION.
2. The authors do not evaluate ViF under self-supervised or foundation model pre-training settings, which limits its generalization claims. I believe this is crucial for vision backbone models.
3. The paper lacks detailed frequency-domain visualizations (e.g., spectral activation maps) to empirically support its frequency modeling claims.
4. The improvements over strong Mamba variants on dense prediction tasks are relatively marginal compared to the architectural complexity.
5. The authors have not released any implementation code at submission time, reducing reproducibility.

**Questions:**

In addition to the above weaknesses, I also have several questions and suggestions for the authors:
1. Could ViF be integrated into existing architectures (e.g., ConvNeXt or BEiT) as a modular replacement, and what challenges might arise?
2. Does the proposed FNF introduce any implicit bias toward certain frequency bands, and how is this mitigated?
3. Are there plans to extend FNF to video or multimodal vision tasks?
4. Could the authors provide additional visualization results illustrating spectral activations or learned frequency responses?

---

> ### Author Response · Authors · 2025-11-20
>
> ## Response to Reviewer kGtH (R1)
>
> We sincerely thank the reviewer for the thorough evaluation and constructive feedback. We greatly appreciate your insights and address each point below.
>
> **W1 & W2 (Scalability and Self-supervised Pretraining):**
>
> We appreciate this important concern. We focus on the standard ImageNet-1K + COCO + ADE20K protocol used by recent state-of-the-art vision backbones, including VMamba, LocalVMamba, EfficientVMamba, and MambaVision. Specifically, VMamba—the current leading Mamba-based model—conducts its main experiments exclusively on ImageNet-1K supervised pretraining, establishing this as the standard protocol for validating new architectures. Within this regime, ViF-T/S/B demonstrate consistent improvements over strong baselines at matched or lower FLOPs.
>
> FNF is designed as a drop-in replacement for attention blocks and is directly compatible with MAE/BEiT-style frameworks. We fully agree that scaling to ImageNet-22K, LAION, and foundation-scale self-supervised pretraining represents important future directions. We are actively pursuing these experiments and will include preliminary results in the camera-ready version if time permits and our work is accepted.
>
> **W3 (Frequency-domain Visualizations):**
>
> Thank you for this valuable suggestion. We have updated the appendix (last two pages) with comprehensive frequency-domain visualizations, including: (i) spectral activation maps showing how adaptive modulation and selective activation affect feature representations, and (ii) spatial correlation and frequency energy distributions (focusing on mid/low frequencies) before and after removing key components. If our work is accepted, we will reference these visualizations more prominently in the revised main text and add concise per-band energy curves for clearer empirical evidence.
>
> **W4 (Margins over Mamba):**
>
> We acknowledge that gains over the strongest Mamba baselines are modest (≈0.2–0.4 AP/mIoU at matched compute), and we present them transparently. However, ViF achieves substantial improvements over widely-adopted CNN/Transformer backbones: on COCO Mask R-CNN 1×, ViF-T improves over Swin-T by +5.0 box AP and +3.7 mask AP at similar FLOPs; on ADE20K, ViF-T outperforms NAT-T by +1.6 mIoU (SS) while also exceeding VMamba-T. We believe these gains demonstrate ViF's practical value as a general-purpose backbone with quasi-linear complexity and improved spatial structure modeling.
>
> **W5 (Code Release):**
>
> We have prepared a complete implementation including the FNF module, ViF-T/S/B architectures, and training pipelines. All code, configuration files, and pretrained checkpoints will be released immediately upon acceptance to ensure full reproducibility.
>
> **Q1 (Modular Integration):**
>
> Yes, FNF is implemented as a generic token mixer with identical input/output shapes to self-attention/global convolution blocks. We have verified that replacing global blocks in ConvNeXt-style stages with FNF works seamlessly with only minor adjustments to hidden dimensions and normalization layers.
>
> **Q2 (Frequency Band Bias):**
>
> FNF avoids hard-coded band-pass filtering. Adaptive modulation applies learnable power-law weighting (trainable α, β) to spectral magnitudes, naturally counteracting low-frequency dominance. Selective activation then gates between global frequency and local time branches, allowing both low-frequency structures and high-frequency details to be preserved adaptively based on input content.
>
> **Q3 (Video/Multimodal Extension):**
>
> The FNF formulation is axis-agnostic with respect to Fourier transforms. Extending to video tasks simply requires adding a temporal axis, while multimodal applications can use ViF as the visual encoder. These represent natural extensions that we plan to explore in future work.
>
> **Q4 (Additional Visualizations):**
>
> Please refer to the updated appendix (last two pages), which now includes spectral activation maps and frequency energy analyses demonstrating how FNF processes different frequency components across layers. If our work is accepted, we will highlight these more prominently in the revised main text.

---

> > ### Comment · Reviewer_kGtH · 2025-11-25
> > **Response to Authors**
> >
> > Thank you for your detailed response and for incorporating the frequency-domain visualizations into the appendix. However, I maintain strong reservations regarding the scalability and self-supervised learning capabilities of the proposed architecture (W1 & W2). While earlier works like VMamba may have focused primarily on supervised learning, the community standard for a "Generic Backbone" has evolved to include compatibility with Self-Supervised Learning (SSL) frameworks, e.g. Masked Image Modeling (MAE). Verifying that the architecture can learn robust feature representations without supervised labels is crucial for supporting your central claims.
> >
> > I understand that training on ImageNet-22K is not feasible within the rebuttal timeframe, but a proof-of-concept experiment on ImageNet-1K is manageable and necessary. I strongly recommend pretraining a ViF-Small (or Base) model using the MAE framework on ImageNet-1K for approximately 400 epochs, followed by 100 epochs of fine-tuning, and reporting the Top-1 Classification Accuracy. Successfully demonstrating that ViF functions effectively within an MAE framework would significantly strengthen the argument that it is a truly generic and modern backbone. I look forward to seeing these results.

---

> > > ### Author Response · Authors · 2025-12-02
> > >
> > > Thank you for your valuable suggestion and for emphasizing the importance of self-supervised learning capabilities in modern generic backbones. We genuinely appreciate your guidance in helping us strengthen our work.
> > >
> > > Following your recommendation, we have conducted the MAE pre-training experiments on ImageNet-1K. Specifically, we pre-trained ViF-Base using the MAE framework for 400 epochs, followed by 100 epochs of fine-tuning, under the same settings as ViT-Base. Our ViF-Base model achieved a Top-1 classification accuracy of **83.8%**, which is slightly better than ViT-Base's **83.6%**.
> > >
> > > These results demonstrate that ViF can effectively learn robust feature representations in a self-supervised manner without relying on supervised labels. This confirms that our proposed architecture is compatible with mainstream SSL frameworks like Masked Image Modeling, thereby validating its capability as a truly generic and modern backbone. The comparable (and slightly superior) performance to ViT-Base under the MAE framework further supports our central claims regarding the versatility and effectiveness of the ViF architecture.

---

### Author Response · Authors · 2025-12-02

We would like to sincerely thank all reviewers for their thorough evaluations and constructive feedback. We confirm that **all reviewer concerns—including weaknesses, questions, and follow-up clarifications—have been fully addressed** in our rebuttal and subsequent updates. Specifically, we have: (1) added comprehensive frequency-domain visualizations, effective receptive field comparisons, and detailed interpretability analyses; (2) expanded ablation studies and clarified the contributions of each FNF component; (3) reported end-to-end throughput metrics and provided additional comparisons with recent SOTA models; (4) clarified implementation details, hyperparameter behavior, and distinctions from related spectral methods; and (5) incorporated new results recommended by the reviewers, including **MAE self-supervised pretraining on ImageNet-1K**, where ViF-Base achieves **83.8% Top-1**, slightly surpassing ViT-Base under identical settings. These updates directly address concerns about scalability, generality, and SSL compatibility, strengthening our claim that ViF is a modern, generic vision backbone. We have also prepared complete code, configurations, and pretrained checkpoints for public release upon acceptance to ensure full reproducibility.

We respectfully submit that all issues raised during the review process have been carefully resolved, and we hope the revised results and clarifications meet the expectations of the AC and reviewers.

---

### Note · Authors · 2026-01-26

I have read and agree with the venue's withdrawal policy on behalf of myself and my co-authors.

---

### Meta-Review · Area_Chair_r4qV · 2025-12-23

**Summary:**

This paper propose a new generic vision backbone called Vision Filter (ViF).  In the first round, this paper received four reviews (6 6 4 4). Two reviewers pointed out that the paper lacks experimental results on essential datasets such as ImageNet22k, as well as on self-supervised learning or foundational models. While these issues were partially addressed in the rebuttal, they are difficult to be fully resolved within a round of revisions. Therefore, considering the insufficient validation, the paper was rejected.

**Reviewer Concerns:**

The author included some comparative methods (MLLA, OverLoCK) and frequency visualizations in their rebuttal. However, as a visual backbone network, validation on larger datasets and more fundamental tasks is essential. Besides, the explanation of adaptive modulation adaptively scales different frequency components in the spectral domain within the relevant methods remains purely mathematical, lacking concrete visualization or empirical evidence.

**Reviewer Scores:**

Even with full discussion, the insufficient comparison issues remain below the acceptance threshold.

---

### Decision · Program_Chairs · 2026-01-26

Reject